# Inhibitory Effect of *Sargassum fusiforme* and Its Components on Replication of Respiratory Syncytial Virus In Vitro and In Vivo

**DOI:** 10.3390/v13040548

**Published:** 2021-03-25

**Authors:** Kiramage Chathuranga, Asela Weerawardhana, Niranjan Dodantenna, Lakmal Ranathunga, Won-Kyung Cho, Jin Yeul Ma, Jong-Soo Lee

**Affiliations:** 1College of Veterinary Medicine, Chungnam National University, Daejeon 34134, Korea; chathurangakiramage@gmail.com (K.C.); aselasampath2009@gmail.com (A.W.); niranjan3k@gmail.com (N.D.); lakmalranathunga13@gmail.com (L.R.); 2Korean Medicine Application Center, Korea Institute of Oriental Medicine, Daegu 41062, Korea; wkcho@kiom.re.kr (W.-K.C.); jyma@kiom.re.kr (J.Y.M.)

**Keywords:** *Sargassum fusiforme*, RSV, therapeutic effects, eicosane, docosane, tetracosane

## Abstract

*Sargassum fusiforme*, a plant used as a medicine and food, is regarded as a marine vegetable and health supplement to improve life expectancy. Here, we demonstrate that *S. fusiforme* extract (SFE) has antiviral effects against respiratory syncytial virus (RSV) in vitro and in vivo mouse model. Treatment of HEp2 cells with a non-cytotoxic concentration of SFE significantly reduced RSV replication, RSV-induced cell death, RSV gene transcription, RSV protein synthesis, and syncytium formation. Moreover, oral inoculation of SFE significantly improved RSV clearance from the lungs of BALB/c mice. Interestingly, the phenolic compounds eicosane, docosane, and tetracosane were identified as active components of SFE. Treatment with a non-cytotoxic concentration of these three components elicited similar antiviral effects against RSV infection as SFE in vitro. Together, these results suggest that SFE and its potential components are a promising natural antiviral agent candidate against RSV infection.

## 1. Introduction

Respiratory syncytial virus (RSV) was discovered in 1956 in the respiratory system of captive chimpanzees [1,2,3]; however, clinical descriptions of RSV with lower respiratory tract infections in infants and elderly people were recorded as early as the 1840s, when these infections were called “congestive catarrhal fever” [1,2]. RSV is a major cause of acute respiratory tract infections and leads to pneumonia and bronchiolitis in infants and toddlers around the world [4,5]. More than 97% of children younger than 5 years old have been infected with RSV at least once [6]. Individuals can be re-infected throughout their life because RSV infection does not induce permanent natural immunity. Consequently, the other most affected age group besides children is elderly people with compromised immunity who can get secondary infections, which lead to severe morbidity and mortality [4]. The health care costs associated with the hospitalization and treatment of RSV-infected patients are a significant economic burden. Studies of the economic impact of RSV-related illnesses suggest that the annual cost of RSV treatment was $18 million in Canada in 1993 and $652 million in the USA in 2000, with the latter cost only including treatment of children younger than 5 years old [7,8].

There are no available efficient RSV vaccines or new antiviral drugs. In the 1960s, formalin-inactivated vaccine administration to the children resulted in vaccine-enhanced disease following the natural RSV infection [9,10,11]. Following this vaccine failure, more studies have been conducted to develop a reliable vaccine candidate for the RSV infection. However, previous attempts to develop a vaccine failed due to an inability to elicit long-lasting and persistent protection against RSV [4]. Antiviral drugs for routine use are not commercially available. Many studies have investigated the use of high titers of RSV immunoglobulin, ribavirin, and corticosteroids against RSV infection [5,12]. However, none of these drugs is convenient and cost-effective for practical use. Therefore, new therapeutics against RSV infection must be investigated due to the high number of infections, the high morbidity and mortality in infants and elderly people, the lack of permanent natural immunity, the unavailability of efficient vaccines, and the huge economic burden caused by RSV infection.

Fossil evidence suggests that folk or traditional medicine, which often employs plants as remedies, was used by Neanderthals approximately 60,000 years ago [13]. According to the World Health Organization, more than 65% of the world’s population currently uses natural components for medicinal purposes [5]. All around the world, the use of natural herbs as food preservatives, supplements, and flavorings has attracted significant interest. Among the thousands of plants and their constituents, those with a long history of consumption as food and traditional medicines should be well studied to investigate their analytical and biological properties [14]. Brown algae are utilized in the human food processing and nutraceutical industries due to their unique bioactive constituents, such as fucoidan, alginate, phlorotannins, and fucoxanthin [14,15]. *Sargassum fusiforme* is an edible brown alga widely distributed around the coastlines of China, Korea, and Japan, and this nutritious marine vegetable has been applied as a therapeutic in traditional Chinese medicine for thousands of years [16]. It is a popular functional seaweed that can prolong life expectancy [17,18]. Modern pharmacological studies demonstrated that *S. fusiforme* is rich in polysaccharides, proteins, and microelements that elicit many beneficial effects in the human body, such as antioxidant effect [19], immunomodulatory effect [20,21], anti-tumor effect [22,23], anti-aging effect [22], hypoglycemic effect [24], anticoagulant effect [25], and anti-inflammatory effect [26]. Moreover, infection and replication of HIV were inhibited by the extract of *S. fusiforme* in T cells, macrophages, and microglia via inhibiting entry and post-entry related steps in the HIV life cycle [27]. A separate study demonstrates that biologically active molecules in *S. fusiforme* could interact with the CD4 receptor and inhibit the fusion of HIV-1; together they proposed *S. fusiforme* as a lead candidate in anti-HIV-1 drug development [28].

This study evaluated the antiviral effects of *S. fusiforme* extract (SFE) against RSV infection in a cell culture model and an in vivo mouse model. We also identified active compounds in SFE, namely, eicosane, docosane, and tetracosane, and confirmed their anti-RSV effects.

## 2. Materials and Methods

### 2.1. Cell Culture, Virus Propagation, and Plaque Assay

Human epithelial type 2: HEp2 (ATCC CCL-23) cells were cultured in dulbecco’s modified eagle medium (DMEM) (Invitrogen, Waltham, MA, USA) supplemented with 10% FBS (Hyclone, Vitoria, Australia) and 1% antibiotic-antimycotic (Gibco, Waltham, MA, USA) (10% DMEM). All cells were kept at 37 °C under 5% CO_2_ in a humid incubator. The green fluorescence protein (GFP)-expressing respiratory syncytial virus was kindly gifted by Dr. Jae U. Jung, Department of Molecular Microbiology and Immunology, University of Southern California, USA. GFP-tagged RSV (RSV–GFP) was propagated in the HEp2 cells and titered by plaque assay [29,30].

### 2.2. Plant Materials and Total Aqueous Extract Preparation

Herb extract was prepared from *Sargassum fusiforme* plant materials obtained from Jaecheon Oriental Herbal Market and the quality of the plant material was verified by Professor Ki-Hwan Bae at the College of Pharmacy, Chungnam National University. The water-soluble extract of *S. fusiforme* was prepared by the Herbal Medicine Improvement Research Center, Korea Institute of Oriental Medicine, Daejeon, Republic of Korea. In the process of preparing water-soluble herbal extract, 100 g of dried plant materials were mixed with distilled water (1L) and heated for 2.5 h at 105 °C using a medical heating plate (Gyeongseo Extractor Cosmos-600, Incheon, Korea) to obtain the extract. The acquired extract was subjected to filtration using a filter paper (0.45 μm, Millex^®^, Darmstadt, Germany) and stored at 4 °C for 24 h. The filtered product was then centrifuged at 12,000 rpm for 15 min, and the supernatant was collected. The pH of the supernatant was adjusted to 7.0 and following pH adjustment total successive aqueous extract was subjected to membrane syringe filtration (0.22 μm) (Millex^®^, Darmstadt, Germany) and stored at −20 °C for further use.

### 2.3. Reagents, Chemicals, and Antibodies

Eicosane, docosane, dotriacontane, tritetracontane, heptacosane, and tetracosane were purchased from Sigma, Missouri, USA. Trypan blue solution was purchased from Glibco (Waltham, MA, USA). Cell cytotoxicity assay kit was purchased from Dojindo Molecular Technologies, INC (CK04: Cell Counting Kit-8, Tokyo, Japan). Antibodies used in the immunoblotting study includes Anti-RSV-G (Abcam, #ab94966, Cambridge, UK), β-actin (Santa Cruz, SC 4777, Dallas, TX, USA), horseradish peroxidase (HRP)-conjugated anti-rabbit IgG (Cell signaling technology, 7074P2, Danvers, MA, USA), HRP-conjugated anti-mouse IgG (Gene Tex, GTX213111-01, Taichung, Taiwan).

### 2.4. Antiviral Assays and Effective Concentration (EC50) Determination

HEp2 cells were used to perform RSV–GFP replication assay as previously described [29] with minor modifications. Briefly, a monolayer of HEp2 cells at about 80% confluency in a 12-well plate was infected with RSV–GFP (multiplicity of infection 0.1). Cells were incubated with viral inoculum in 1% FBS containing DMEM for 2 h at 37 °C. After incubation, cells were washed with phosphate-buffered saline (PBS), and the medium was replaced by 10% FBS containing DMEM and treated with indicated concentrations of SFE, eicosane, docosane, dotriacontane, tritetracontane, heptacosane, or tetracosane. At 48 h post-infection (hpi), GFP expression was measured with Glomax multi-detection system (Promega, Madison, WI, USA) following the manufacturer’s directions. RSV–GFP titer in supernatant and cells was determined by plaque assay [30] and cell viability was determined via trypan blue exclusion test, as previously described [31]. To determine the effective concentration (EC_50_), HEp2 cells were seeded in a 24-well plate; ultimately, EC_50_ values were calculated as the extract concentration yielding 50% GFP expression.

### 2.5. Cell Viability Assay

The CC_50_ value of SF extract and chemicals was determined using Cell counting kit-8 (Dojindo Molecular Technologies, Rockville, MD, USA), as previously described [32]. Briefly, HEp2 cells were treated with indicated concentrations of SFE or chemicals. At 48 h post-treatment, cells were incubated for 1 h with cell counting kit-8 (CCK-8) solution (10 μL/well) in a 37 °C-incubator. Absorbance was measured at 450 nm using a microplate reader (Molecular devices) after the incubation.

### 2.6. Quantitative RT-PCR (qRT PCR)

A monolayer of HEp2 cells was infected with 0.1 multiplicity of infection (MOI) of RSV–GFP or kept uninfected. Infected cells were treated with SFE or chemical compounds at 2 hpi. Cells were collected at indicated time points and kept at −70 °C until further use. Total RNA was extracted from cells or 1 g of lung homogenate (see Section 2.9) using the RNeasy Mini kit (Qiagen, Hiden, Germany), and RNA was reverse transcribed into cDNA using reverse transcriptase (Toyobo, Japan), as described in the manufacturer’s instructions. Quantitative real-time polymerase chain reaction (qRT PCR) was performed on the Rotor Gene Q instrument (Qiagen, Hiden, Germany) using SYBR Green PCR Master Mix (Qiagen, Seoul, Korea). The transcription level of mRNA was obtained by the 2^−∆∆Ct^ method as described previously [33] and expressed as fold induction. The fold induction was analyzed and compared to non-infected control mice sample or non-infected control cell sample. The RT-PCR primer sequences as follows: RSV-G forward primer 5′-CCAAACAAACCCAATAATGATTT-3′ reverse primer 5′-GCCCAGCAGGTTGGATTGT-3′ Glyceraldehyde 3-phosphate dehydrogenase (GAPDH): forward primer 5′-TGACCACAGTCCATGCCATC-3′ reverse primer 5′-GACGGACACATTGGGGG TAG-3′.

### 2.7. Immunoblot Analysis

A monolayer of HEp2 cells was cultured in six-well plates and was infected with 0.1 MOI RSV–GFP. Cells were incubated with viral inoculum in 1% FBS containing DMEM for 2 h at 37 °C. After incubation, the medium was replaced by 10% FBS containing DMEM and treated with SFE or chemicals. The cells were harvested at 0, 8, 12, and 24 hours post treatment. The cell pellets were washed with PBS and subjected to immunoblot analysis. Briefly, harvested cells were lysed in lysis buffer containing 1% NP-40, 150 mM NaCl, 50 mM Tris-HCl pH 8.0, and a protease inhibitor. The lysed samples were centrifuged to remove the cell debris. Cell lysates were mixed with 10x sample buffer (Sigma, St. Louis, MI, USA) at a 1:1 ratio and denatured at 100 °C for 10 min. Total proteins were separated by sodium dodecyl sulfate-polyacrylamide gel electrophoresis (SDS–PAGE). Separated proteins were transferred into a polyvinylidene fluoride (PVDF) membrane (Bio-Rad, Hercules, CA, USA) in buffer containing 30 mM Tris, 200 mM glycine, and 20% methanol for 2 h. The membrane was blocked in 5% bovine serum albumin (BSA, Sigma) in Tris-buffered saline + Tween 20 (TBST) for one hour and probed with anti-RSV-G antibody or anti-β-actin antibody with 5% BSA in TBST. Proteins were detected by incubating with a secondary anti-rabbit IgG-HRP or anti-mouse IgG-HRP for 1 h. the HRP reaction was visualized with the enhanced chemiluminescence detection system (ECL-GE Healthcare, Pittsburgh, PA, USA) using a Las-3000 mini Lumino Image Analyzer (GE Life science, Pittsburgh, PA, USA). To analyze the band intensities of Western blot images, ImageQuant LT 7.0 (GE-Life Science, Pittsburgh, PA, USA) software was used. Β-actin band intensity was used to normalize the RSV-G protein expression.

### 2.8. Syncytial Formation and Syncytia Counting Assay

The ability of the SFE to block cell-to-cell spread was evaluated using GFP expression in the cells. A monolayer of Hep2 cells at about 80% confluency in a 12-well plate was infected with RSV–GFP (0.1 MOI). Cells were incubated with viral inoculum in 1% FBS containing DMEM for 2 h at 37 °C. After incubation, the medium was replaced by 10% FBS containing DMEM and treated with SFE. After 48 h, incubation cells were washed with cold PBS, and cell images were taken under 400 magnifications. ImageJ software was used to quantify the number of syncytia. Untreated RSV-infected monolayer was used as control.

### 2.9. RSV–GFP Challenge Experiment in Mouse Modal

After three days of acclimation under experimental conditions, five-week-old BALB/c mice were separated into experimental groups as virus-infected and SFE-treated group (*n* = 6), virus-only group (*n* = 6), and uninfected group (*n* = 2). Mice were anesthetized using ketamine prior to intranasal infection of RSV–GFP 1 × 10^6^ PFU (28 μL per head). Mice were orally administered with 200 μL of 0.5 mg/mL SFE at 6, 12, 18, and 24 hpi. Lung tissues from euthanized mice were collected aseptically at five-day post-infection (dpi). Lung RSV titration was determined by RSV-G protein mRNA transcription fold quantification. RSV-G protein mRNA level was quantified as described above. In vivo experiments included in this study were approved by the Institutional Animal Care and Use Committee of Chungnam National University (Approval number CNU-00816 and approval date: 27 September 2016).

### 2.10. Statistical Analysis

Statistical analysis was performed using GraphPad Prism software version 6 for Windows (GraphPad Software, San Diego, CA, USA). All the data were from at least of three independent experiments, and data are shown as means ± standard deviations. Comparison between herb or chemical treated groups with untreated control groups were analyzed by the unpaired t-test. In all experiments, * *p* < 0.05, ** *p* < 0.01 or *** *p* < 0.001 was regarded as statistically significant.

## 3. Results

### 3.1. Anti-RSV Effect of SFE

Aqueous extracts from an herb library consisting of 200 herb extracts were screened to detect antiviral activity against RSV [5] and SFE was selected as a positive candidate against RSV infection inhibition. The Hep2 cell line, derived from human carcinoma, is a commonly used cell type to investigate the effect of RSV. First, the ability of different concentrations of SFE to inhibit the replication of GFP-tagged RSV (RSV–GFP) was investigated. RSV–GFP expression, RSV titer, and RSV-induced cell death were evaluated in Hep2 cells treated with different concentrations of SFE (30, 50, or 100 μg/mL). Hep2 cells treated with all three concentrations of SFE exhibit significant inhibition of GFP expression, compared to untreated control against RSV–GFP infection (Figure 1A,B). On the other hand, SFE significantly suppresses RSV titer in infected Hep2 cells in a dose-dependent manner (Figure 1C). Cell viability assay indicates that the treated doses of SFE were able to significantly inhibit the RSV-induced cell death at 48 h post-infection (hpi) (Figure 1D). These results demonstrate that SFE significantly reduced RSV replication in Hep2 cells. Treatment with 50 μg/mL SFE most effectively inhibited viral replication and virus-induced cytotoxicity, and therefore, this concentration was used in subsequent in vitro experiments.

### 3.2. Determination of the Effective Concentration (EC_50_) and Cytotoxic Concentration (CC_50_) of SFE

The effective concentration (EC_50_) of SFE against RSV in HEp2 cells was determined by performing a virus yield inhibition assay, as aforementioned [27]. Briefly, the yield of GFP expression was examined, and the depletion of 50% virus titer was taken as equivalent to a 50% reduction in GFP expression. Figure 2A indicates that SFE inhibits 50% of RSV–GFP infection (0.1 MOI) at a concentration of 45.34 μg/mL. Thereafter, we evaluated the cell cytotoxic concentration (CC_50_) of SFE based on cell cytotoxicity assay using HEp2 cells. CC_50_ value obtained for SFE was 659.24 μg/mL (Figure 2B), which is greater than 80% cell viability. The selectivity index (SI) indicates the safety of a crude extract against RSV infection [34]. The SI of SFE was 14.53 (Figure 2C). These data suggest that SFE is safe to use as a therapeutic agent against RSV infection.

### 3.3. Therapeutic Effect of SFE against RSV Infection

To assess the therapeutic effect of SFE on virus replication, GFP absorbance was measured at given time points to check the virus replication after SFE treatment. Treatment with 50 μg/mL SFE significantly reduced GFP expression and the RSV titer at 36 and 48 hpi, but not at 12 or 24 hpi (Figure 2D,E). Next, a virus replication assay was performed to assess the ability of SFE to inhibit RSV replication at different time points after virus infection. A monolayer of HEp2 cells was infected with RSV–GFP at 0.1 MOI and SFE treatment was performed at given specific time points, thereafter GFP absorbance was measured at 48 hpi. Results indicate increased GFP expression as the interval between virus infection and SFE treatment increases (Figure 2F,G). Comparably, results of plaque assay indicate higher virus titer in the sample where SFE was treated at 24 hpi and low virus titer in the samples of which SFE treatment was carried out in early time points after RSV–GFP infection (Figure 2H).

### 3.4. Effect of SFE on Viral Protein Synthesis and Viral RNA Expression

The ability of SFE to inhibit intracellular viral RNA transcription and protein translation was evaluated in HEp2 cells. A monolayer of HEp2 cells was infected with RSV–GFP (0.1MOI) and was treated with 50 μg/mL SFE at 2 hpi. Then, cells were harvested at 0, 12, 24, and 36 hpi. Viral mRNA expression was determined by qRT-PCR, and viral protein synthesis was evaluated by immunoblot analysis. According to Figure 3A, SFE reduces RSV-G protein synthesis in HEp2 cells. To validate the antiviral effect observed with viral protein synthesis reduction by SFE treatment, the RSV-G protein mRNA transcription level was examined next. As shown in the figure, mRNA fold induction of 40.78 ± 0.52 and 124.35 ± 1.22 in the virus-only group could reduce up to 19.295 ± 0.275 and 78.18 ± 1.34 with SFE treatment at 24 hpi and 36 hpi, respectively (Figure 3B). These results demonstrate that the SFE-mediated reduction of RSV gene transcription and protein synthesis are associated with the antiviral activity of SFE in vitro.

### 3.5. Effect of SFE on Syncytium Formation

RSV-mediated syncytium formation is a distinguished phenomenon underlying cell-to-cell infection that has a significant impact on in vitro virus spreading. A syncytium formation assay with infected HEp2 cells was performed to determine whether SFE prevents the cell-to-cell spread of RSV after infection. HEp2 cells were infected with RSV–GFP and untreated or treated with SFE (50 μg/mL) at 2 hpi. Syncytium formation was examined at 48 hpi. Large areas of syncytium formation were visible in untreated HEp2 cells. By contrast, syncytium formation was significantly reduced in SFE-treated HEp2 cells at 48 hpi (Figure 3C). Furthermore, the number of syncytial was counted using ImageJ software, as described previously [3]. SFE reduced the number of syncytial in HEp2 cells in a dose-dependent manner (Figure 3D).

### 3.6. Oral Administration of SFE Protects against RSV Infection in BALB/C Mice

In accordance with the guidelines of the Institutional Animal Care and Use Committee of Chungnam National University, a mouse model was adopted for in vivo study to assess the therapeutic effects of SFE against RSV infection. BALB/c mice were intranasally infected with RSV–GFP (1 × 10^6^ PFU/mouse) or left uninfected and orally administered phosphate-buffered saline (PBS) or SFE at 6, 12, 18, and 24 hpi. Results of preliminary studies were used to select the optimum virus infection dose and inoculation time. Mice were euthanized 5 days post-inoculation and the lungs were collected. Lung homogenate was used to determine the RSV-G mRNA level by qRT-PCR. The RSV-G mRNA level in the respiratory tract at 5 dpi shows a marked reduction in SFE-treated mice than in PBS-treated mice (Figure 3E). These data demonstrate that SFE protects against RSV infection in vivo by inhibiting the virus replication.

### 3.7. Non-Cytotoxic Concentrations of Tetracosane, Docosane, and Eicosane Inhibit RSV Replication In Vitro

Previous reports identified several active components of SFE. Among these, eicosane, dotriacontane, tritetracontane, docosane, heptacosane, and tetracosane are major components of SFE and have been proposed to have antimicrobial activity [35]. To assess the antiviral effect of selected components, HEp2 cells were infected with RSV–GFP and treated with 100 mM of each component individually at 2 hpi, after which virus replication was monitored. GFP expression was significantly lower in HEp2 cells treated with tetracosane, docosane, and eicosane than in untreated HEp2 cells (Figure 4A). Three selected components from the screening were confirmed for the anti-RSV effect (Figure 4B,C). The results were similar to those presented in Figure 4A. Moreover, a standard plaque formation assay was used to determine the virus titer. Results show a similar trend as the GFP expression analysis where selected chemicals markedly inhibit RSV titer (Figure 4D). The EC_50_ of eicosane, docosane, and tetracosane in HEp2 cells was 86.41 ± 2.36 mM, 71.25 ± 4.23 mM, and 74.95 ± 1.25 mM, respectively, while the CC_50_ of eicosane, docosane, and tetracosane in HEp2 cells was 314.24 ± 8.25 mM, 389.25 ± 7.14 mM, 250.04 ± 1.27 mM, respectively (Figure 4E–G). The SI of tetracosane, docosane, and eicosane against RSV–GFP was 3.65, 5.44, and 3.33, respectively (Figure 4H). Furthermore, treatment with tetracosane, docosane, and eicosane reduced RSV-G protein synthesis in HEp2 cells (Figure 5A–C). This reduction in viral protein translation was associated with viral gene transcription. Treatment with these three components individually significantly reduced the RSV-G mRNA level at 24 and 36 hpi (Figure 5D). These in vitro data demonstrate that tetracosane, docosane, and eicosane inhibit RSV replication in HEp2 cells.

Next, the synergistic anti-RSV effect of these three active components was determined in HEp2 cells. HEp2 cells were infected with RSV–GFP (0.1MOI) for 2 h in Dulbecco’s modified Eagle medium (DMEM) containing 1% fetal bovine serum (FBS), and then treated with tetracosane, docosane, and eicosane separately or in combination (1:1:1) in DMEM containing 10% FBS. GFP absorbance was measured, and the virus titer was determined by the standard plaque assay at 48 hpi. Treatment with tetracosane, docosane, or eicosane separately or in combination significantly reduced GFP expression (Figure 5E). However, the effects of treatment with a combination of these three active components did not significantly differ from the effects of treatment with these components individually at the same final concentration. A similar result was obtained when virus replication was quantified by the standard plaque assay (Figure 5F). These data demonstrate that tetracosane, docosane, and eicosane do not exhibit synergistically enhanced anti-RSV activity.

## 4. Discussion

Respiratory tract-related viral diseases can lead to severe illness at all ages of the lifetime. Unfortunately, most viral infections of the respiratory tract cannot be prevented, and therefore, effective therapeutics with disease-limiting ability are urgently required. Thousands of people around the world have relied on terrestrial and marine medicinal plants for primary care in history and present-day [36]. Traditional Marine Chinese Medicines (TMCMs) have been recognized as harmless, natural convincing therapeutic agents for thousands of human complications and are extensively used individually or in combination due to their high efficacies and fewer side effects [37]; therefore, it is important to study the biological functions and chemical compositions of natural compounds used as TMCMs [38]. Seaweed is one of the indispensable TMCMs, which was documented in the oldest Materia medica book titled *Shennong’s Classic of Materia Medica* thousands of years ago [39] and has been predominantly applied in different prescriptions. *S. fusiforme* is one of only two marine plants with a seaweed origin recorded in the Pharmacopoeia of China [40] and is widely distributed in Korea, China, and Japan. Pharmacological studies have demonstrated that several beneficial effects of the edible brown algae *S. fusiforme* [41,42,43,44], which has been used as a traditional medicine to treat scrofula, beriberi, and thyroid diseases for thousands of years [22,42,45,46].

Thousands of antiviral agents derived from natural sources with high efficacies of virus clearance and low toxicities to the host are considered to be promising. Up-to-date number of these natural herbal products with antiviral properties have been wildly used all around the world [34]. The present study identified *S. fusiforme* extract (SFE) with a potent anti-RSV effect by screening a library comprising more than 200 herbs [5]. SFE showed promising dose-dependent antiviral effects against RSV infection in HEp2 cells (Figure 1), and the CC_50_ value of SFE was several magnitudes higher than its EC_50_ value (Figure 2), which is consistent with an admiring safety profile. Moreover, oral administration of SFE significantly reduced RSV gene transcription in the lungs of infected mice (Figure 3). Furthermore, we investigated that chemical constituents of SFE, namely, tetracosane, docosane, and eicosane, were involved in its activity against RSV infection (Figure 4 and Figure 5).

Treatment with excessively high concentrations of medicinal herbs causes cytotoxicity and complicates clinical trials [47]. Therefore, we examined the cytotoxicity of SFE in HEp2 cells; SFE did not show to induce cytotoxicity in HEp2 cells. The CC_50_ of SFE was 659.24 µg/mL, which is several magnitudes higher than its EC_50_ of 45.34 µg/mL. The SI of 14.53 indicates that SFE has a wide safety margin for therapeutic clinical purposes. In the inhibition assay of RSV replication, we investigated RSV gene transcription and RSV-G protein synthesis in infected HEp2 cells. The RSV-G mRNA level was significantly reduced in SFE-treated HEp2 cells at 24 and 36 hpi (Figure 3B), and RSV-G protein synthesis was also significantly reduced in SFE-treated HEp2 cells at late time points after virus infection (Figure 3A). These reductions in viral mRNA transcription and protein synthesis positively correlated with a low level of viral replication (Figure 2D,E). Syncytium formation by RSV remarkably contributes to virus spread in vivo facilitating cell-to-cell virus infection [48]. SFE treatment also significantly reduced RSV-induced syncytium formation in HEp2 cells (Figure 3C,D). Since SFE could inhibit RSV replication in vitro, we continued to test the antiviral potential in vivo. In vivo replication of RSV can be accurately assessed by qRT-PCR [49].

A positive interconnection was found between viral replication and clinical RSV symptoms during natural and experimental infections [50,51]. Therefore, control of viral replication in the lung is an important parameter for assessing protection against RSV. In vivo experiment, oral administration of SFE significantly reduced RSV-G mRNA transcription in the lungs of RSV-infected mice at 5 dpi and these results are harmonious with the low virus replications observed in HEp2 cells treated with SFE in vitro. The exact mechanism underlying the anti-RSV effects of SFE is still being investigated in vitro and in vivo. However, it is possible that a significant blockade of RSV replication or cell-to-cell RSV infection by specific components of SFE ultimately inhibits virus spreading from infected sites to neighboring sites, which might improve viral clearance from the lungs of SFE-treated mice.

Polysaccharides, fatty acids, phytosterols, phlorotannins, and meroterpenoids are considered to be the major chemical components of *Sargassum* [52,53]. Recent studies have shown that *S. fusiforme* contains significantly high amounts of alkane (eicosane, tetracosane, dotriacontane, and tritetracontane) and fatty acid natured hydrocarbons (docosane and heptacosane) [35]. In our preliminary studies, we identified eicosane, docosane, and tetracosane as active components of SFE for potent anti-RSV effects. These compounds have been reported with numerous biological properties. Eicosane for anti-fungal, anti-bacterial, and anti-tumor activities [46]; docosane for anti-bacterial activity [54]; and tetracosane for anti-bacterial and anti-tumor activities [55]. Therefore, we evaluated the anti-RSV effect of these three compounds, and interestingly, all three chemicals reduced RSV replication similar to SFE in HEp2 cells. However, the EC_50_ value was over 50 mM in the tested three active components. The anti-RSV effect of SFE could be due to other unknown active chemical components together with eicosane, docosane, and tetracosane. Furthermore, the SIs of these active components against RSV in vitro indicate that they have wide safety margins for therapeutic applications (Figure 4H); however, the values were lower than that of the SFE (Figure 2C). Furthermore, they did not elicit synergistic anti-RSV effects compared with their individual effects at the same concentrations (Figure 5E,F). Further detailed studies are also needed to elucidate the mechanism underlying the anti-RSV effects of these active chemical constituents.

In summary, we investigated SFE as a potential antiviral candidate that exhibits an anti-RSV effect in cultured cells and in vivo mouse model, and eicosane, docosane, and tetracosane from SFE are also identified in this study for their apparent anti-RSV effects. Our results suggest that SFE and its potential components could be a promising antiviral agent against RSV infection, and consumption of SFE would have beneficial effects in the prevention or therapy of RSV infection.

## Figures and Tables

**Figure 1 viruses-13-00548-f001:**
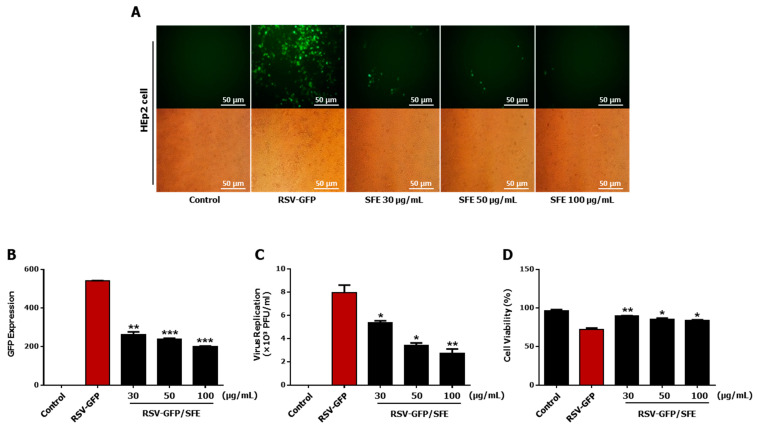
Antiviral activity of *Sargassum fusiforme* extract (SFE) in HEp2 cells. HEp2 cells were seeded into 12-well cell culture plates with the cell number of 1.5 × 10^5^ cells/well. After 12 h, the medium was changed to 1% fetal bovine serum (FBS) containing Dulbecco’s modified Eagle medium (DMEM), and cells were infected with Green Florescence protein tagged Respiratory Syncytial Virus (RSV–GFP) 0.1 Multiplicity of Infection(0.1 MOI) or kept uninfected. After 2 h, the medium was replaced with 10% FBS containing DMEM, and cells were treated with 30, 50, 100 (μg/mL) *Sargassum fusiforme* extract (SFE). Cells without any treatment regard as virus only. (**A**) After 48 h, images were obtained (200 × magnification). (**B**) Green Fluorescence protein (GFP) absorbance levels were measured by Glomax multi-detection luminometer (Promega). (**C**) Viruses were titrated from the cell supernatant and cells by standard plaque assay. (**D**) Cell viability was determined by trypan blue exclusion assay at 48 hpi. Scale bar-50 μM. GFP absorbance, cell viability, and virus titer expressed as mean ± SD. Error bars indicate the range of values obtained from counting duplicates in three independent experiments (* *p* < 0.05, ** *p* < 0.01, and *** *p* < 0.001 regarded as significant difference).

**Figure 2 viruses-13-00548-f002:**
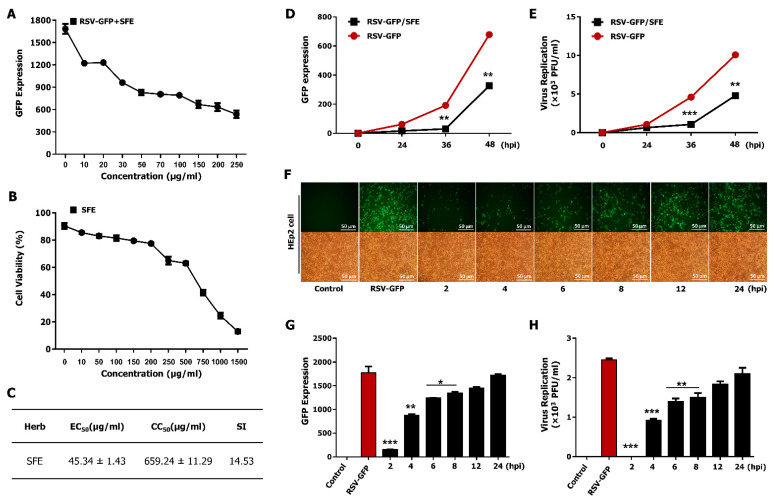
The therapeutic effect of SFE against RSV–GFP infection. (**A**) A monolayer of HEp2 cells in a 12 well plate was infected with RSV–GFP (0.1MOI) for 2 h, and the medium was changed into 10% FBS containing DMEM with indicated concentrations of SEF and GFP expression levels were measured at 48 hpi. (**B**) The viability of HEp2 cell upon SFE treatment with indicated concentrations was measured at 48 hpi. (**C**)The ratio between CC_50_ and EC_50_ was considered as selectivity index (SI). A monolayer of HEp2 cells cultured in a 12-well plate was infected with RSV–GFP (0.1MOI) for 2 h. (**D**) GFP expression levels at different time points after 50 μg/mL SFE treatment, and (**E**) viral load in the cells and cell supernatant was measured by standard plaque assay, (**F**–**H**) 50 μg/mL SFE treatment to the infected cells was performed at indicated time points following virus infection. (**F**) GFP expression images. (**G**) GFP expression levels. (**H**) Virus titer from both cells and cell supernatant were taken at 48 hpi. Scale bar-50 μM. GFP absorbance and virus titer expressed as mean ± SD. Data are representative of three independent experiments, each with similar results (* *p* < 0.05, ** *p* < 0.01 and *** *p* < 0.001 regarded as significant difference).

**Figure 3 viruses-13-00548-f003:**
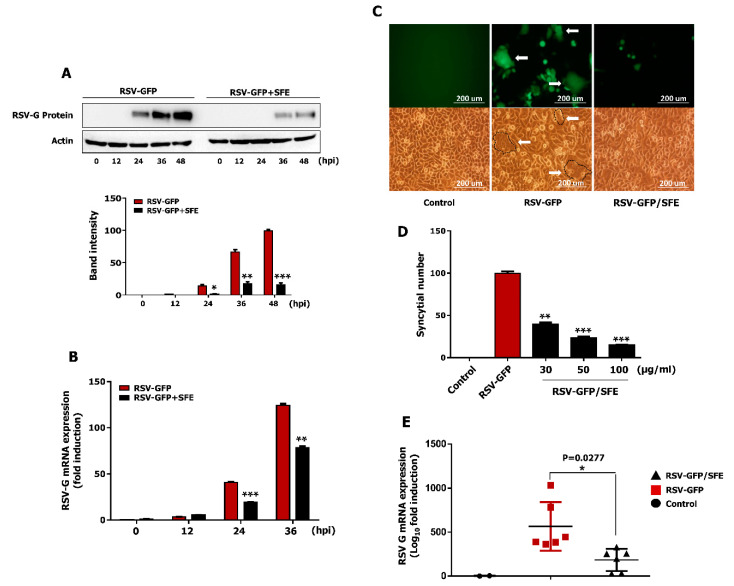
SFE abrogates viral gene transcription, protein translation, and syncytial formation in HEp2 cells and reduces RSV–GFP replication in mice model. Hep2 cells (3 × 10^5^ cell/well) were added to the six-well cell culture plate. After 12 h, a monolayer of HEp2 cells was infected with 0.1 MOI of RSV–GFP in 1% FBS containing media for 2 h. Next, cells were treated with 50 μg/mL of SFE. (**A**) RSV-G protein expression level was evaluated by immunoblot analysis assay with the cell lysates collected at indicated time points. RSV-G protein intensity was determined compared to β-actin. (**B**) Transcription levels of RSV-G protein at indicated time points were evaluated by qRT-PCR analysis. GAPDH was used for the normalization. (**C**) Cell and GFP images were taken at 48 hpi to see the syncytial formation inhibition by SFE (400× magnification), white arrow: cell to cell fusion and syncytial formation in HEp2 cells. (**D**) The syncytial number was counted using image software. (**E**) Five-weeks-old BALB/c mice (RSV–GFP, *n* = 6, RSV–GFP/SFE, *n* = 6, Control, *n* = 2,) were intranasally infected with RSV–GFP (1 × 10^6^ PFU/mice) in a total volume of 28 μL. SFE were orally administrated with the dose of 200 μL/mice (0.5 mg/mL) at 6, 12, 18, and 24 hpi. The transcription level of RSV-G protein mRNA in the lung tissues of the mice at 5 dpi was determined by qRT-PCR. White arrow: RSV-syncytial formation. Scale bar-200 Μm. Western blot band intensity, mRNA transcription, syncytial number was expressed as the mean ± SD of three independent experiments. In vivo experiments were conducted twice. (* *p* < 0.05, ** *p* < 0.01, *** *p* < 0.001 regarded as significant difference).

**Figure 4 viruses-13-00548-f004:**
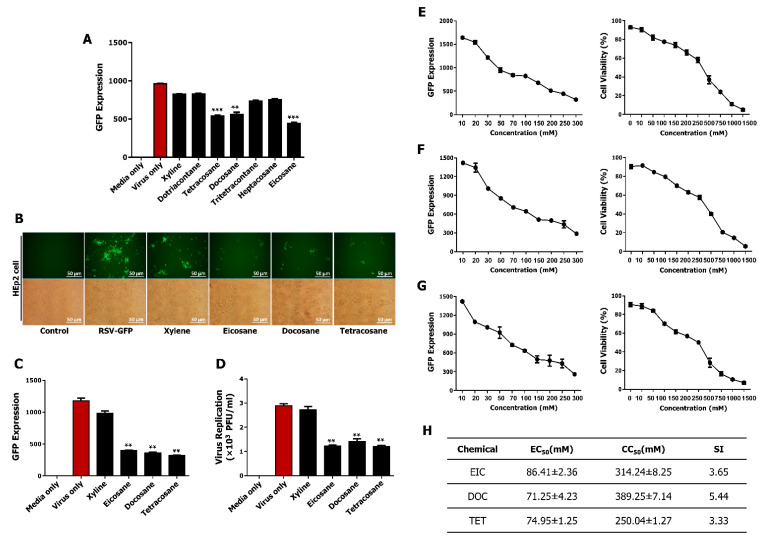
Identification and antiviral effect of eicosane (EIC), docosane (DOC) and tetracosane (TET) in vitro. A monolayer of HEp2 cells was infected with RSV–GFP (0.1 MOI) for 2 h with DMEM containing 1% FBS. Then, the medium was replaced with DMEM containing 10% FBS, and cells were treated with 100 mM of given chemicals. (**A**) GFP absorbance levels were measured by Glomax multi-detection luminometer (Promega). HEp2 cells were infected with 0.1 MOI of RSV–GFP and 2 hpi cells were treated with 100 mM of EIC, DOC, and TET. (**B**) Cell images were obtained (200 × magnification) with a fluorescence microscope. (**C**) Fluorescence absorbance level was determined with Glomax multi-detection luminometer. (**D**) RSV–GFP titration was performed by standard plaque assay. (**E**–**G** left panels) The amount of chemical required for reduction of 50% GFP absorbance value was considered as EC_50_ value. (**E**–**G** right panels) CC_50_ values of the three chemicals were determined. Cell viability was determined by Cell counting kit-8 assay. (**H**) Selectivity index (SI) CC_50_/EC_50_. Scale bar-50 μM. Results indicate the values obtained from counting duplicates in three independent experiments (** *p* < 0.01 *** *p* < 0.001 regarded as significant difference).

**Figure 5 viruses-13-00548-f005:**
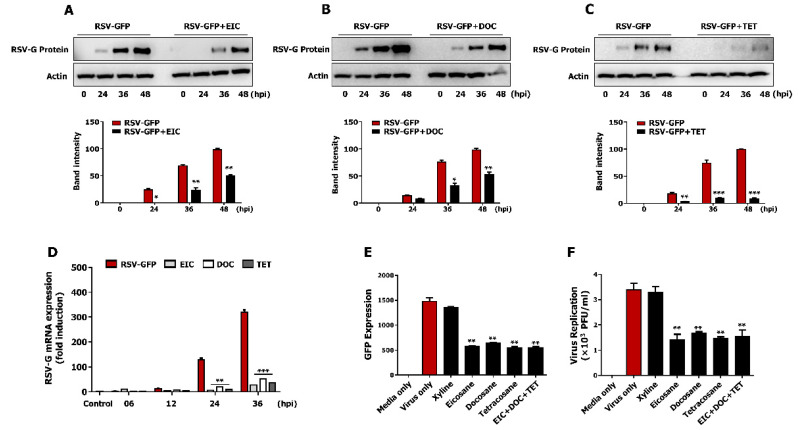
Antiviral effect of eicosane (EIC), docosane (DOC) and tetracosane (TET) in vitro. RSV–GFP infected cells were treated with 100 mM of EIC, DOC, and TET at 2 hpi, cells were harvested at indicated time points. (**A**–**C**) RSV-G protein expression was determined by immunoblotting with an anti-RSV-G protein antibody, and the intensity of the RSV-G was quantified. (**D**) RSV-G protein mRNA transcription level was determined by qRT-PCR. GAPDH was used for normalization. (**E**,**F**) Infected cells were treated with 100 mM concentration of EIC, DOC, TET, and a combination of EIC, DOC, and TET at 2 hpi and (**E**) GFP absorbance levels were measured by Glomax multi-detection luminometer (Promega) after 48 hpi. (**F**) Viruses were titrated from the cell supernatant and cells by standard plaque assay. GFP absorbance and virus titer expressed as mean ± SD. Error bars indicate the range of values obtained from counting duplicates in three independent experiments (* *p* < 0.05 ** *p* < 0.01 *** *p* < 0.001 regarded as significant difference).

## Data Availability

The data that support the findings of this study are available from the corresponding author upon reasonable request.

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
