# Peer review of "Inhibitory Effect of Sargassum fusiforme and Its Components on Replication of Respiratory Syncytial Virus In Vitro and In Vivo"

_viruses, 2021, doi:10.3390/v13040548_

Round 1
Reviewer 1 Report
Chathuranga et al. in this manuscript investigate the impact of S. fusiforme extract on the viral replication of RSV. They employ a combination of in vitro and in vivo experiments to identify a number of compounds found in S. fusiforme which appear to demonstrate an antiviral effect.
Major Comments:
- Introduction contains several inaccuracies and/or unsupported claims. In addition, more background regarding the role of fusiforme in therapeutic applications of viral infections should be included.
- The authors need to show that the reductions in RSV-GFP activity being shown in the figures are due to SFE inactivation of the virus and not due to cell death caused by toxicity. Other studies (Chakrabarty et al., 2011) have shown that organic extracts in high concentrations can induce apoptosis in human carcinoma cells lines like HeLa cells (which Hep2 cells are derived from). It is very likely that a large impact of these extracts on reduction of virus is simply due to cell death rather than viral inhibition. The authors here show in Figure 2G that the cell viability drops significantly with increased concentrations of extract and the authors fail to separate how the 20 – 30% reduction in cell viability at 50 – 100 ug/ml from the impacts on virus. In addition, the authors report that the CC50 is 659.24 ug/ml, but the data in the figure supports a far lower value closer to 200 ug/ml. The authors need to clarify or correct. The manuscript would be improved by experiments being performed in non-cancerous cells lines or cells less susceptible to toxicity caused by SFE.
- The mouse data shown (see Figure 3E) would be greatly improved by showing more data including reductions in viral load via plaque assay. Using only G transcript expression, it is unclear what specifically may be causing this single point of reduction.
Minor Comments:
- RSV was first characterized in 1956 in chimpanzees, not 1963. (see line 29)
- The authors mention that previous attempts at a vaccine failed due to inability to elicit adequate protection, however, it should also be noted that early vaccines triggered vaccine-enhanced disease which resulted in a significant setback in the field for vaccine discovery. (see lines 43 – 45).
- The authors claim that fusiforme can prolong life expectancy but fail to provide scientific evidence to support this claim. They need to provide a reference for this statement. (see line 65)
- Please correct the spelling error of “titter” with “titer” throughout.
- There is a paragraph split mid-sentence that needs removed. (see line 211)
- The images in the figures need to be enlarged, they are far too small and the quality of the images is figures is limiting.
Author Response
Dear Reviewer,
We would like to thank you for evaluating our work and we are pleased to have an opportunity to revise and improve the manuscript. We thoughtfully considered the comments and revised the manuscript in accordance with the suggestions and now we believe that the manuscript is greatly improved.
Thank you very much.
Sincerely yours,
Jong-Soo Lee

Reviewer 2 Report
Dear Authors,
Your paper entitled "Inhibitory effect of Sargassum fusiforme and its components on replication of Respiratory Syncytial Virus in vitro and in vivo" is a very interesting study.
Sargassum fusiforme, a plant used as a medicine and food, is regarded as a marine vegetable and health supplement to improve life expectancy.
You have demonstrated that S. fusiforme extract (SFE) has antiviral effects against Respiratory Syncytial Virus (RSV) in vitro and in vivo mouse model. Treatment of HEp2 cells with a non-cytotoxic concentration of SFE significantly reduced RSV replication, RSV-induced cell death, RSV gene transcription, RSV protein synthesis and the syncytium formation. Moreover, oral inoculation of SFE significantly improved RSV clearance from the lungs of BALB/c mice. Interestingly, the phenolic compounds eicosane, docosane, and tetracosane were identified as active components of SFE. Treatment with a non-cytotoxic concentration of these three components elicited similar antiviral effects against RSV infection as SFE in vitro. Together, these results suggest that SFE and its potential components are a promising natural antiviral agent candidate against RSV infection. I recommend publishing this study.
I suggest to repeat this research in case of SARS-CoV-2 infection. Hopefully SFE will be effective too.
Author Response
Dear Reviewer,
We would like to thank you for evaluating our manuscript and giving us a important suggestion to improve our future research work. We deeply appreciate your effort on evaluating our manuscript.
Thank you very much.
Sincerely yours,
Jong-Soo Lee

Reviewer 3 Report
In the present manuscript from Chathuranga et al., the authors study the antiviral activity against respiratory syncytial (RSV) virus of S. fusiforme extracts. Although the authors suggest that S. fusiforme extracts could inhibit RSV replication, the study presents lot of unclear data. Most of lot, the antiviral effects are observed only for high levels of compounds, that cannot be considered for efficient antiviral treatment.
Material and methods :
1/ Some paragraph such as 2.4 (antiviral assay, L107-118), 2.5 (infectivity inhibition assay, L119-127), and 2.6 (cell viability assay, L 128-134) present redundancies and should be revised.
2/ Paragraph 2.7 Quantitative RT-PCR (L135-145) must be completed to clearly indicate the treatment of RT-PCR data (standardization).
Results
1/ Authors initiated their study by showing the antiviral effect of SFE on infected Hep2 cells. They first show the effect on RSV infection using 30, 50 and 100 µg/ml of SFE extracts and decided to further investigate the effect of SFE using the dose of 50µg/ml (L204-205). However, fig 2 F and G (that are not “therapeutic effect” but “antiviral effect” based on the legend), showing viral infection and toxicity at different doses of SFE, should have been integrated in the first figure1. It is noteworthy that the quality of Fig 1A should be improved, and scale bars indicated.
The scale of x axis of the graph 2 F and G is surprising (0, 10, 20, 30, 50, 70, 100, 200 and 250 µg/ml and 0, 10, 50, 100, 150, 200, 250, 500, 750, 1000, and 1500 µg/ml?). Based on these graphs, EC50 seems close to 100µg/ml and CC50 close to 200µg/ml, whereas authors indicated 45µg/ml and 659 µg/ml respectively in table 2H and in the text. In conclusion, the all study is poorly convincing.
2/ Fig 2A : same comment than for Fig 1A.
3/ Fig3 A : how was performed the quantification?
4/ Based on figure 3B the authors wrote that SFE reduced transcription of RSV-G mRNA by 21 and 46-fold at 24 and 36 hpi, respectively (L264-265) : these results are really not clear based on this figure.
5/ Fig3E : Given the toxicity of SFE extracts ad the four serial administration to mice, authors should have include in vivo toxicity data. Of not, considering the control infected group, only two animals displayed strong infection, which induce a bias in the results : the antiviral effect seems quite limited if considering the 4 other animals. Finally, authors indicated n=14 in the legend of the figure which is confusing as 2 mice are not infected. This should be specified.
6/ Fig 4: The authors tested the effect of EIC, DOC, and TET on RSV infection using huge amounts of molecules (mg/ml). Such concentrations are really high compared to concentration of SFE extracts tested, and the fact that no synergy or additive effect was observed is surprising and poorly convincing. For fig 4B, same comments than for Fig1A and 2A. For figure 4E, F, and G : same comments than for graphs 2 F and G. Of note, the legend indicate that cells were treated with 100 mg/ml of compounds, but the value 200 mg/ml is indicated in the text (L328).
7/ Conclusion : Overall, the authors over interpreted their data on the efficiency of SFE and should have considered the doses tested and the toxicity. They emphasized that SFE inhibits RSV replication not only at the transcriptional level but also at the post-transcriptional level (L410-411): globally, if mRNA are downregulated, it is not surprising that viral proteins are less expressed, and thus that less syncytia can be observed (as they are depend on viral F expression). These observations only confirm the effect on mRNA transcription. The discussion should be deeply revised.
8/ General comment : English should be revised by an English speaker.
Author Response

(The authors gave the same response as above.)

Round 2
Reviewer 1 Report
The authors have addressed most of my issues with this revision. There are a few typos and/or grammatical issues needing addressed (please perform a read-through for issues), however the manuscript is greatly improved.
Minor comments:
Line 77 - "demonstrate" should be "demonstrated"
Line 249 - "results of plaque assay" should be "results of the plaque assay"
Author Response
Dear reviewer,
We deeply appreciate each comment and suggestion from the reviewer. Our manuscript greatly improved with the reviewers' comments at the first round of the revision. Here, we revised the manuscript and made every effort to address comments to the best of our possibility.
Thank you.
Jong-Soo Lee
Responses to the Reviewers’ comments (comments from the reviewer in black, responses to the reviewer in blue)
Reviewers' comments:
Reviewer #1 (Comments for the Author):
Comment)
The authors have addressed most of my issues with this revision. There are a few typos and/or grammatical issues needing addressed (please perform a read-through for issues), however the manuscript is greatly improved.
Response)
We thank the reviewer for evaluating our work. Comments at the first round of the revision from the reviewer were greatly helpful to improve our manuscript. Here we have carefully went through the manuscript and edited our previous mistakes in grammer and typos according to reviewers comments. We would like to thank the reviewer again for pointing out important points and helping us to improve our manuscript.
Minor comments:
Comment)
Line 77 - "demonstrate" should be "demonstrated"
Response)
We thank the reviewer for showing the mistake and we corrected the point. (Line number: 78)
Comment)
Line 249 - "results of plaque assay" should be "results of the plaque assay"
Response)
We thank the reviewer for showing the mistake and we corrected the point. (Line number: 252)
Reviewer 3 Report
The authors have responded the comments previously adressed. The figures and the text of the manuscript have been revised accordingly.
However, proofreading should be done to avoid redundancies (especially in figures' legends and in the discussion) and to improve the english langage.
Author Response
Dear reviewer,
We deeply appreciate each comment and suggestion from the reviewer. Our manuscript greatly improved with the reviewers' comments at the first round of the revision. Here, we revised the manuscript and made every effort to address comments to the best of our possibility.
Thank you.
Jong-Soo Lee
Responses to the Reviewers’ comments (comments from the reviewer in black, responses to the reviewer in blue)
Reviewers' comments:
Reviewer #3 (Comments for the Author):
Comment)
The authors have responded the comments previously adressed. The figures and the text of the manuscript have been revised accordingly. However, proofreading should be done to avoid redundancies (especially in figures' legends and in the discussion) and to improve the english langage.
Response)
We thank the reviewer for evaluating our work. Previous comments from the reviewer were greatly helpful to improve our manuscript. Here we have carefully went through the manuscript and edited our previous mistakes in grammer and typos according to reviewers comments. Moreover, we removed some repetitive words from the figure ledgends. We would like to thank the reviewer again for pointing out important points and helping us to improve our manuscript.